# Temporal trends in heart failure medication prescription in a population-based cohort study

Alicia Uijl [1,2,3,4] Ilonca Vaartjes,[1] S Denaxas,[3,4,5,6] Harry Hemingway [3,4,6] Anoop Shah,[3,4] J Cleland,[7,8] Diederick Grobbee,[1] Arno Hoes,[1] Folkert W Asselbergs,[3,4,9,10] Stefan Koudstaal[3,4,9]

For numbered affiliations see end of article.

**Correspondence to**
Dr Alicia Uijl;
a.uijl@umcutrecht.nl

## ABSTRACT

**Objective** We examined temporal heart failure (HF) prescription patterns in a large representative sample of real-world patients in the UK, using electronic health records (EHR).

**Methods** From primary and secondary care EHR, we identified 85 732 patients with a HF diagnosis between 2002 and 2015. Almost 50% of patients with HF were women and the median age was 79.1 (IQR 70.2–85.7) years, with age at diagnosis increasing over time.

**Results** We found several trends in pharmacological HF management, including increased beta blocker prescriptions over time (29% in 2002–2005 and 54% in 2013–2015), which was not observed for mineralocorticoid receptor-antagonists (MR-antagonists) (18% in 2002–2005 and 18% in 2013–2015); higher prescription rates of loop diuretics in women and elderly patients together with lower prescription rates of angiotensin-converting enzyme inhibitors and/or angiotensin II receptor blockers, beta blockers or MR-antagonists in these patients; little change in medication prescription rates occurred after 6 months of HF diagnosis and, finally, patients hospitalised for HF who had no recorded follow-up in primary care had considerably lower prescription rates compared with patients with a HF diagnosis in primary care with or without HF hospitalisation.

**Conclusion** In the general population, the use of MR-antagonists for HF remained low and did not change throughout 13 years of follow-up. For most patients, few changes were seen in pharmacological management of HF in the 6 months following diagnosis.

## INTRODUCTION

Heart failure (HF) is a common public health burden, with the prevalence of HF estimated at approximately 500 000 patients in the UK.[1 2] Once diagnosed, initiation and up titration of guideline recommended therapies can reduce morbidity and mortality, although 5-year survival still remains 20%–50%.[3 4]

Several observational studies have assessed treatment uptake in patients with HF following their diagnosis. These studies suggest that many patients did not receive guideline-recommended therapies or at low

### Strengths and limitations of this study

► Large cohort of patients with heart failure (HF) from primary and secondary care.
► Long follow-up period of almost 15 years.
► Unable to differentiate between HF subphenotypes such as HF with reduced, mid-range or preserved ejection fraction.
► Unknown treatment eligibility, contraindications or intolerances may affect the choice of treatment.

doses with sparse attempts for up titration.[5–8] Optimal treatment for effective disease management seems to be particularly challenging in elderly patients, women or patients with multiple comorbidities and contraindications for treatments.[7 8] At present, few data are available for prescription trends in patients with HF in the general population and even fewer data are available that shed light on medication use in patients with HF in the years prior to their HF diagnosis.

The CALIBER (CArdiovascular disease research using LInked Bespoke studies and Electronic health Records) resource curates primary and secondary care electronic health records (EHR) of 5 million individuals in the UK, including HF diagnosis and medication prescriptions.[9] Given the amount of information available, medication use of all patients with HF in the community may be investigated—including those which are underrepresented in randomised clinical trials.

Therefore, we sought to examine HF treatment prescription patterns following a HF diagnosis for the overall population as well as specific subgroups based on gender (eg, women), age (eg, elderly), social economic status and healthcare setting (eg, primary care or secondary care), in a large representative sample of real-world patients in the UK, using EHR.[10]

## METHODS

### Data source

Patients were selected from linked EHR in the UK, which consist of three linked databases: The Clinical Practice Research Datalink (CPRD) with primary care EHR, Hospital Episodes Statistics (HES) containing coded diagnoses and surgical procedures from inpatient hospital admissions and the Office for National Statistics registry containing cause-specific mortality data.[10] Previous work has shown that these patients are representative of the general population in the UK.[11 12]

### Study population

Patients were included at their first record of HF from CPRD or HES between 1 January 2002 and 31 December 2015. In CPRD, events were defined by a diagnosis of HF based on Read (V.2) controlled clinical terminology codes (National Health Service-coded clinical terms) and in HES by a diagnosis of HF based on International Classification of Diseases 10th revision (ICD-10) codes. The same HF diagnosis codes were used as in previous papers, with in addition several newer Read codes listed in online supplemental table 1.[4 13] All patients were eligible for inclusion if they were aged 18 years or older and were registered with a GP for at least 1 year prior to diagnosis of HF, in a practice that had at least 1 year of up-to-standard data recording in CPRD (data quality check). The first record of HF from CPRD or HES was considered the index date. Individuals were censored at the earliest date from the date of deregistration in CPRD, the last data collection date of a practice in CPRD, the date of death or at the study end date (31 December 2015). Data on EHR phenotyping variables from patients with HF up to 3 years prior to index date were included in this study.

### Patient and public involvement

There was no patient or public involvement in this research.

### EHR phenotyping variables

Baseline patient characteristics were based on records from CPRD and/or HES prior to index date, including demographics (age, sex, ethnicity, social deprivation), cardiovascular risk factors (smoking, body mass index, diastolic blood pressure and systolic blood pressure and estimated glomerular filtration rate), comorbidities (a medical history of atrial fibrillation, chronic obstructive pulmonary disease, diabetes, hypertension, ischaemic heart disease, valvular disease and history of cancer) and medication prescription.

CPRD includes all prescriptions from the general practice. Prescriptions in CPRD were classified as RAS inhibitors (angiotensin-converting enzyme inhibitors and/or angiotensin II receptor blockers), beta blockers, mineralocorticoid receptor-antagonists (MR-antagonists) and loop diuretics. Definitions of these variables could be found online at http://www.caliberresearch.org/portal/.

Medication prescription for RAS inhibitors, beta blockers, MR-antagonists and loop diuretics was identified between 3 years prior to HF diagnosis up to 3 years after HF diagnosis per the following increments: −36 months to −24 months, −24 months to −18 months, −18 months to −12 months, −12 months to −6 months, −6 months to −3 months, −3 months to HF diagnosis, HF diagnosis to +3 months, +3 months to +6 months, +6 months to +12 months, +12 months to +18 months, +18 months to +24 months and +24 to +36 months.

Healthcare setting was characterised as primary care only (no HF hospitalisation), secondary care only (no Read HF diagnosis recorded in primary care) or HF diagnosis in both primary and secondary care. Ethnicity records from CPRD and HES were combined and categorised as Caucasian, Asian, Black or Other. Social deprivation was measured as quintiles of the index of multiple deprivation of the geographical area of the primary care practice, a score was calculated based on seven indices of deprivation: income, employment, health and disability, education, barrier to housing and services, crime and living environment.[14] Smoking status in CPRD was classified as never, ex-smokers or current smokers.

### Statistical analysis

Patient characteristics were summarised as mean (SD) or median (IQR) for continuous variables and percentages for categorical variables. The percentage of patients with HF prescribed pharmacological treatments was calculated per increment and per time period as defined by publication year of previous European Society of Cardiology (ESC) guidelines (2001, 2005, 2008 and 2012)[1 15–18]: 2002–2005, 2006–2008, 2009–2012 and 2013–2015. In addition to the overall cohort, we investigated several subgroups: age (<vs≥ 75 years old), sex (men vs women), social economic status (lowest quintile of social deprivation vs the rest) and setting (only follow-up in primary care vs only in secondary care vs follow-up in primary care after HF hospitalisation). All analyses were performed using R V.3.6.1.

## RESULTS

### Baseline characteristics

We identified 85 732 patients with a HF diagnosis. The study flow diagram is found in online supplemental figure 1. Median follow-up after HF diagnosis (index date) was 2.1 years (0.6–4.5 years). Table 1 shows the overall baseline patient characteristics and per time period 2002–2005, 2006–2008, 2009–2012 and 2013–2015. Almost 50% of patients were women and the median age was 79.1 (70.2–85.7) years, with age at HF diagnosis increasing over time. Overall, many patients with HF had comorbidities, most common were hypertension (61%), ischaemic heart disease (44%) and atrial fibrillation (37%), with increasing numbers of patients with comorbidities over time. Approximately 40% (n=34 489) of patients were followed-up in primary care after a HF hospitalisation,

**Table 1** Patients characteristics of patients with heart failure between 2002 and 2015

| | Overall | 2002–2005 | 2006–2008 | 2009–2012 | 2013–2015 | % missing |
|---|---|---|---|---|---|---|
| n | 85732 | 25366 | 17715 | 26114 | 16537 | |
| **Demographics** | | | | | | |
| Age (years, median (IQR)) | 79.1(70.2, 85.7) | 78.7(70.7, 84.9) | 78.7(69.9, 85.4) | 79.5(70.1, 86.3) | 79.7(70.0, 86.4) | 0 |
| Sex (% women) | 48.6 | 49.3 | 48.4 | 48.4 | 48.0 | 0 |
| Ethnicity (% Caucasian) | 96.5 | 97.5 | 96.9 | 96.1 | 95.1 | 3.5 |
| Social deprivation* (% lowest quintile) | 24.3 | 25.1 | 25.0 | 24.0 | 22.9 | 0 |
| **Clinical and lifestyle measurements** | | | | | | |
| SBP (mm Hg, mean (SD)) | 136.2 (20.7) | 140.6 (22.3) | 135.9 (20.7) | 134.6 (20.0) | 132.9 (18.7) | 13.0 |
| DBP (mm Hg, mean (SD)) | 76.2 (12.0) | 78.4 (12.0) | 76.2 (12.0) | 75.4 (12.0) | 74.4 (11.6) | 13.0 |
| BMI (kg/m², mean (SD)) | 28.6 (6.6) | 28.2 (6.4) | 28.4 (6.6) | 28.7 (6.8) | 28.8 (6.8) | 54.0 |
| eGFR (min/m²/1.73mL, median (IQR)) | 58.4(45.3, 72.1) | 54.7(43.4, 66.1) | 56.5(44.3, 68.8) | 60.5(46.3, 75.3) | 62.9(47.5, 78.2) | 24.0 |
| Smoking status (% current) | 20.8 | 22.3 | 20.0 | 20.4 | 20.5 | 38.7 |
| **Medical history (%)†‡** | | | | | | |
| Atrial fibrillation | 36.6 | 28.4 | 36.3 | 40.6 | 43.0 | – |
| COPD | 17.9 | 14.8 | 17.3 | 19.5 | 21.0 | – |
| Diabetes | 22.3 | 18.1 | 22.2 | 23.7 | 26.7 | – |
| Hypertension | 60.7 | 46.0 | 60.7 | 67.9 | 72.0 | – |
| Ischaemic heart disease | 44.2 | 39.0 | 46.0 | 46.4 | 46.8 | – |
| Valvular disease | 16.5 | 9.5 | 14.9 | 19.9 | 23.8 | – |
| **Medication prescription up to 3 months after HF diagnosis (%)‡** | | | | | | |
| RAS inhibitors | 60.8 | 59.6 | 63.5 | 62.0 | 57.6 | – |
| Beta blockers | 42.5 | 28.9 | 41.0 | 49.3 | 54.1 | – |
| MR-antagonists | 18.0 | 18.4 | 17.9 | 17.6 | 18.2 | – |
| Loop diuretics | 63.0 | 68.4 | 63.5 | 61.1 | 57.0 | – |

*Assessed by index of multiple deprivation.
†Denotes prior medical history of given comorbidity.
‡Medical conditions and prescriptions were considered absent if not recorded.
BMI, body mass index; COPD, chronic obstructive pulmonary disease; CPRD, clinical practice research datalink; DBP, diastolic blood pressure; eGFR, estimated glomerular filtration rate; MR, mineralocorticoid receptor; RAS inhibitors, angiotensin-converting enzyme inhibitors and/or angiotensin II receptor blockers; SBP, systolic blood pressure.

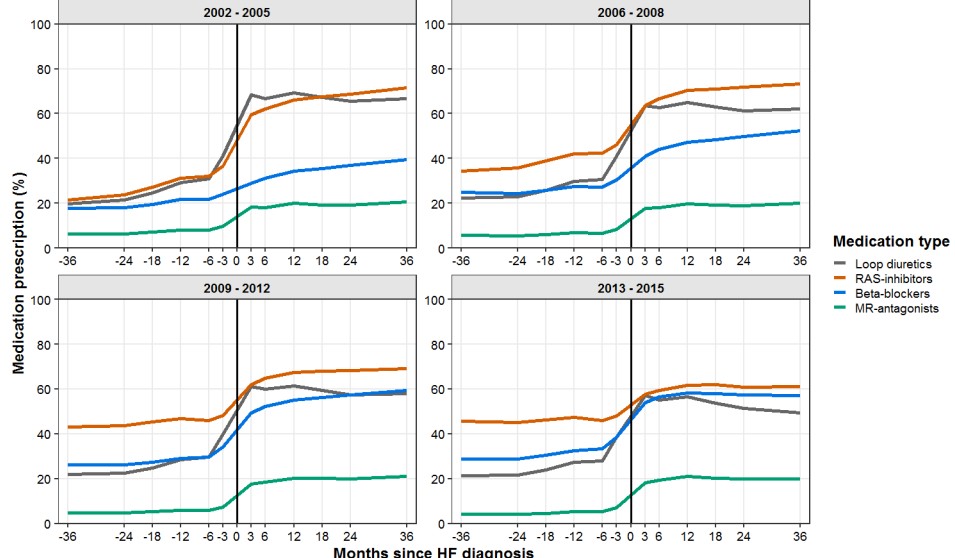

**Figure 1** The percentage of patients with HF receiving prescriptions of RAS inhibitors, beta blockers, MR-antagonists, loop diuretics per months since HF diagnosis. HF, heart failure; MR-antagonists, mineralocorticoid receptor antagonists; RAS inhibitors, angiotensin-converting enzyme inhibitors and/or angiotensin II receptor blockers.

20% (n=15 330) of patients were only known in primary care and never hospitalised for HF and the remaining 40% (n=35 913) of patients had no follow-up in primary care after HF hospitalisation.

### Overall prescription patterns

Overall prescription patterns are shown in figure 1. Many patients were prescribed medication before HF diagnosis, especially RAS inhibitors (20% in 2002–2005 to 46% in 2013–2015). Over time, beta blocker prescription after HF diagnosis increased from 30% in 2002–2005 to 55% in 2013–2015. Throughout the follow-up of 13 years, there were little observed changes for MR-antagonist

uptake, this remained at 20% throughout time after HF diagnosis. The largest observed changes in prescription patterns occurred between 6 months before and after HF diagnosis (figure 1). Approximately 20% of patients with HF were prescribed a loop diuretic up to 3 years prior to HF diagnosis.

### Setting-specific prescription patterns

Setting-specific prescription patterns are shown in figure 2. Patients followed-up in primary care after HF hospitalisation had the highest prescription rates for all types of medication. Over time, the prescription for loop diuretics, RAS inhibitors and beta blockers converged

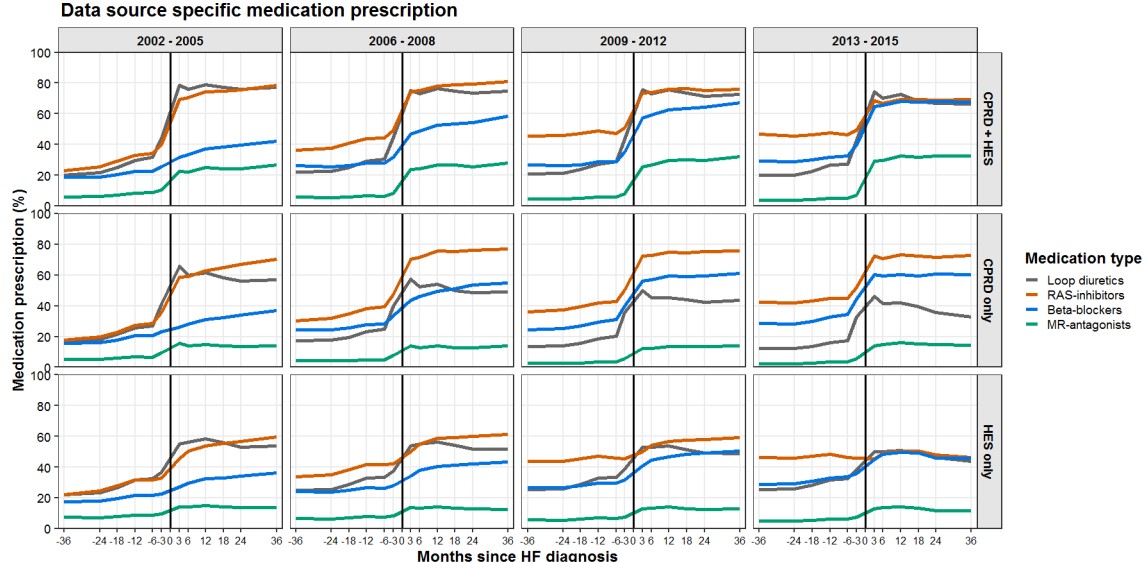

**Figure 2** The percentage of patients with HF receiving prescriptions of RAS inhibitors, beta blockers and MR-antagonists per months since HF diagnosis, stratified by setting (primary care only, secondary care only, both primary and secondary care). HF, heart failure; MR-antagonists, mineralocorticoid receptor antagonists; RAS inhibitors, angiotensin-converting enzyme inhibitors and/or angiotensin II receptor blockers.

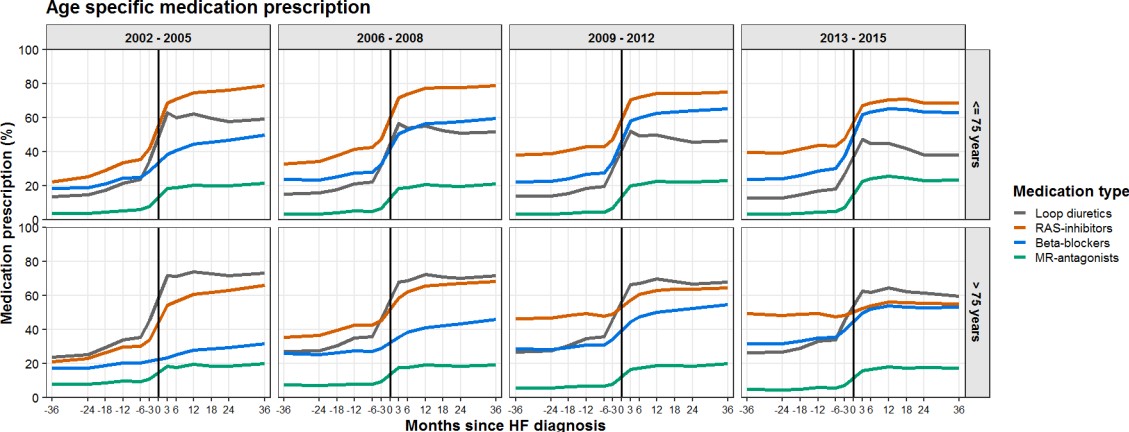

**Figure 3** The percentage of patients with HF receiving prescriptions of RAS inhibitors, beta blockers and MR-antagonists per months since HF diagnosis, stratified by age. HF, heart failure; MR-antagonists, mineralocorticoid receptor antagonists; RAS inhibitors, angiotensin-converting enzyme inhibitors and/or angiotensin II receptor blockers.

together. In these patients, the prescription for MR-antagonists increased over time after HF diagnosis from 20% in 2002–2005 to 30% in 2013–2015.

Patients known in primary care but never hospitalised for HF had lower prescription rates for all types of treatment compared with patients with primary care follow-up and at least one HF hospitalisation. Mainly loop diuretics were less prescribed in these patients and the prescription of loop diuretics decreased over time with 65% of patients receiving loop diuretics after HF diagnosis in 2002–2005 compared with just over 40% in 2013–2015.

Patients hospitalised for HF but without a HF diagnosis in primary care had the lowest prescriptions rates for loop diuretics, RAS inhibitors and beta blockers, which remained stable over time (50%, 45% and 45% in 2013–2015, respectively). MR-antagonists were only prescribed in 13% of patients after HF diagnosis, this was similar for each time period.

### Age-specific prescription patterns

Differences in prescription according to age categories are shown in figure 3. The observed increase in prescriptions for RAS inhibitors, beta blockers and MR-antagonists between 6 months before HF diagnosis and 6 months after HF diagnosis was less pronounced in elderly patients. The average increase in elderly patients was 12%, 7%, 8% for RAS inhibitors, beta blockers and MR-antagonists, respectively, while younger patients had an average increase of 23%, 19% and 13% for RAS inhibitors, beta blockers and MR-antagonists, respectively. On the other hand, a higher proportion of elderly patients was treated with loop diuretics compared with younger patients, both before and after HF diagnosis (45% before and 63% after HF diagnosis in elderly compared with 27% before and 47% after HF diagnosis for younger patients in 2013–2015). After HF diagnosis, a higher percentage of younger patients was prescribed with RAS inhibitors and beta blockers compared with older patients.

### Sex-specific prescription patterns

Differences in prescription between men and women are shown in figure 4. Loop diuretics were prescribed in a higher proportion of women compared with men, this difference was already present prior to HF diagnosis

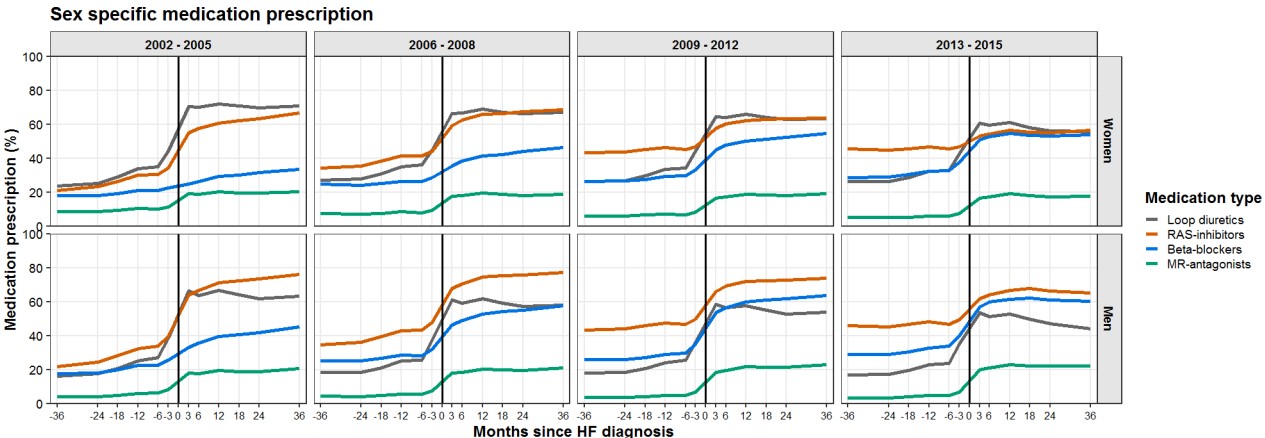

**Figure 4** The percentage of patients with HF receiving prescriptions of RAS inhibitors, beta blockers and MR-antagonists per months since HF diagnosis, stratified by sex. HF, heart failure; MR-antagonists, mineralocorticoid receptor antagonists; RAS inhibitors, angiotensin-converting enzyme inhibitors and/or angiotensin II receptor blockers.

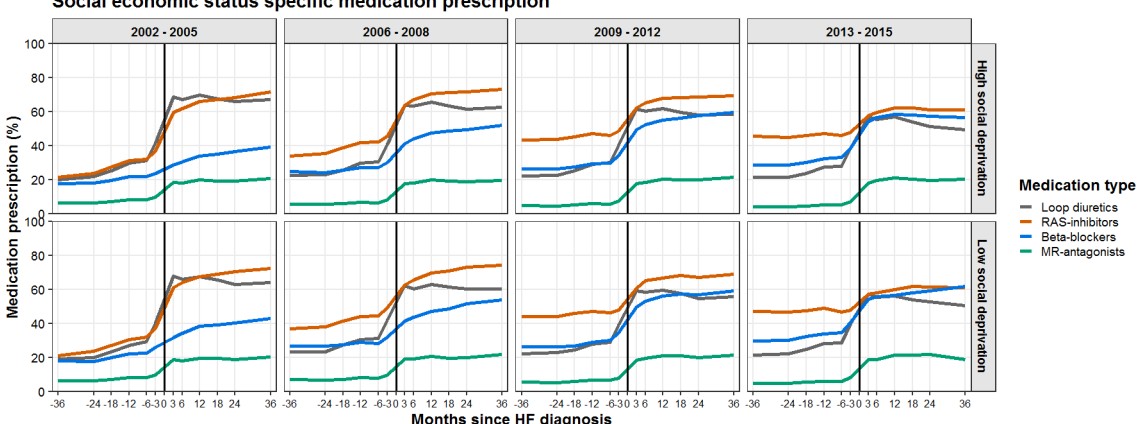

**Figure 5** The percentage of patients with HF receiving prescriptions of RAS inhibitors, beta blockers and MR-antagonists per months since HF diagnosis, stratified by social status (highest quintile of social deprivation vs the rest). HF, heart failure; MR-antagonists, mineralocorticoid receptor antagonists; RAS inhibitors, angiotensin-converting enzyme inhibitors and/or angiotensin II receptor blockers.

where 6 months before diagnosis, 30% of women and 20% of men were prescribed a loop diuretic. After HF diagnosis, the most prescribed medication for women was a loop diuretic, while a higher proportion of men was prescribed a RAS inhibitor. Men were also more often prescribed RAS inhibitors, beta blockers and MR-antagonists after HF diagnosis compared with women.

### Social economic status-specific prescription patterns

Social economic status-specific prescription patterns are shown in figure 5. We did not observe any discernible differences between patients in low versus high social economic areas (highest quintile of social economic deprivation).

### DISCUSSION

In this large-scale study of 85 732 patients with HF, we investigated treatment prescription patterns in a representative sample of real-world patients with HF in the UK between 2002 and 2015. We found three important trends in pharmacological HF management: (a) increased use of beta blockers, whereas there was no increased uptake of MR-antagonists over 13 years follow-up, (b) prescription rates remained almost unchanged after the first 6 months following a HF diagnosis and, finally, (c) higher rates of loop diuretics in women and elderly patients together with lower prescription rates for RAS inhibitors, beta blockers or MR-antagonists.

### Temporal trends in HF medication

Even though prescription rates increased over time from 2002 to 2015, overall prescription rates remained low. This is in line with previously published studies.[5–8 19] Low prescription rates could be attributed to the mixed HF cases found in EHR. We were unable to distinguish HF with reduced ejection fraction (HFrEF), HF with mid-range ejection fraction and HF with preserved ejection fraction (HFpEF) based on medical records, thereby

including known differences in treatment recommendations for these HF phenotypes.[1]

We found no major differences in prescription behaviour after the publication of ESC guidelines; however, we did observe the gradual increase of beta blockers as one of the cornerstones of HF treatment. RAS inhibitors were prescribed in a high proportion of patients throughout the years of the study, presumably because the first clinical trials in HFrEF showing a beneficial effect were from the late 1980s and early 1990s.[20] Surprisingly, we found lower than expected prescription rates for MR-antagonists, which persisted over the years included in this study. This is in spite of multiple clinical trials which have shown benefit in patients with HFrEF.[21] Besides HFrEF trials, a post hoc analysis of the Treatment of Preserved Cardiac Function Heart Failure With an Aldosterone Antagonist (TOPCAT) trial in 2015 (Spironolactone, an MR-antagonist, for HFpEF) reported regional differences between Americas and Russia/Georgia, where the American patients showed clinical benefits.[22] The American College of Cardiology/American Heart Association focused update on HF management in 2017 gave spironolactone a grade IIb recommendation, thereby stimulating that selected patients with HFpEF could be treated with spironolactone to decrease rehospitalisations.[23]

### HF medication initiation following diagnosis

Most activity in treatment prescription behaviour was observed between 6 months before and 6 months after HF diagnosis. After the 6 month mark, we did not observe many patients starting any of the medication investigated. This is in line with previous studies showing that there are few changes in medication use and little up titration of medication after treatment initiation.[5 24] This leaves room for improvement in starting treatment at any time point following a HF diagnosis, for example, if patients hospitalised with acute HF do not immediately tolerate negative inotropic medication such as beta blockers.

## Impact of HF hospitalisation on medication prescription

We found differences in prescription patterns between patients with a HF diagnosis recorded in different settings. Patients with a primary care HF diagnosis without HF hospitalisation had much lower prescription rates of loop diuretics compared with patients with a HF diagnosis recorded in both primary and secondary care. It could be that these patients have less severe fluid overload or symptoms that require alleviation by loop diuretics, and thus less severe HF.

Previously it was shown that there are differences in overall 5-year survival of patients with HF diagnosis recorded in primary care only, secondary care only and in both, with the worst survival seen in patients with HF identified only in secondary care and the best survival for patients with HF identified in primary care with or without hospitalisation for HF.[4] Here, we advance current knowledge by showing that there are longitudinal differences in HF care of patients with diagnosis recorded in different settings.

In this study, almost 40% of patients did not have a GP record of a HF diagnosis after a HF hospitalisation. One reason could be that GPs do not formally register HF with a Read diagnosis code, but rather in free-text descriptions. However, there could also be a potential quality of care gap or failure of communication between secondary and primary care, where patients are not treated optimally. Primary care is the basis of many healthcare systems, including the UK. If there is no HF diagnosis recorded in primary care after HF hospitalisation, which is shown to be indicative for worse survival, rehospitalisation and severity of disease, this could be detrimental for patients.

## HF treatment in women and elderly

Over time, we observed that HF was diagnosed at a later age, with the median almost 80 years old between 2013 and 2015. This is also seen in many other developed countries where the mean age of HF diagnosis is over 70 years old.[25 26]

We observed lower prescription rates in elderly patients compared with younger patients for RAS inhibitors, beta blockers and MR-antagonists, although the difference in MR-antagonists was less pronounced. Many elderly patients were already using RAS inhibitors prior to HF diagnosis, therefore, the increase in prescription rate is not as steep as compared with younger patients with HF who are prescribed less medication prior to HF diagnosis. This could be explained by the presence of comorbidities, such as atrial fibrillation or hypertension, which are much more prevalent among elderly compared with younger patients, and for which these elderly patients could be prescribed RAS inhibitors.

Remarkably, the difference between prescription of RAS inhibitors and beta blockers prior to HF diagnosis was less than 5% for men and women, and only after the diagnosis of HF was a higher proportion of men prescribed a RAS inhibitor or beta blocker. This could potentially be related to the fact that elderly women are more likely to develop HFpEF and, therefore, tend to be treated symptomatically with loop diuretics, rather than with RAS inhibitors and beta blockers. However, literature also shows that there are differences in treatment prescription in men and women with HFrEF, for which there is no obvious explanation.[27]

Both elderly patients and women received more loop diuretics. However, this could potentially be harmful, especially for elderly, since loop diuretics could lead to electrolyte disturbances and acute kidney injury.[28] Elderly patients are often excluded or underrepresented in clinical trials, therefore, current recommendations lack convincing evidence in the elderly population. However, recently a large meta-analysis reported a significant effect of beta blockers on overall mortality regardless of age.[29] These studies indicate that elderly patients also benefit from HF-specific medication and should be a choice of treatment for these patients, besides loop diuretics for symptom alleviation. However, elderly patients might have more contraindications or intolerances to RAS inhibitors, beta blockers and MR-antagonists and might, therefore, be more often treated with loop diuretics for symptom control.

## Strengths and limitations

Strengths of this study are the large cohort of patients with HF and a long follow-up period. Patient records available are representative of the general UK population, which provides evidence for the validity of using these EHR for research.[11 12] However, we were limited by the inability to differentiate between HF phenotypes based on medical records, since there was no access to detailed echocardiography estimates to assess systolic function. Nor did we have information on New York Heart Association class or N-terminal pro–B-type natriuretic peptide biomarker levels. Furthermore, treatments administered during a hospital admission or discharge were not reported, such as intravenous inotropic agents. However, CPRD includes all prescriptions from general practice to non-hospitalised patients. We were also unable to assess patients' symptom class (which would affect their eligibility for treatments such as MR-antagonists) and contraindications or intolerances that may affect the choice of medication.

## CONCLUSION

The results of this population-based study of over 80 000 patients with HF in England show variable increases in uptake of evidence-based treatments, with no change in prescription of MR-antagonists over 13 years but an increase in beta blocker use. Large differences were observed between HF patient groups, with lowest prescription rates of RAS inhibitors, beta blockers and MR-antagonists in women, elderly patients and those without a HF diagnosis in primary care. Most changes in prescriptions occurred within 6 months prior to or 6 months following a diagnosis of HF, with little change, thereafter suggesting further opportunities to improve HF management.

**Author affiliations**
[1]Julius Global Health, Julius Center for Health Sciences and Primary Care, University Medical Center Utrecht, Utrecht University, Utrecht, The Netherlands
[2]Division of Cardiology, Department of Medicine, Karolinska Institutet, Stockholm, Sweden
[3]Institute of Health Informatics, University College London, London, UK
[4]Health Data Research UK, London, UK
[5]The National Institute for Health Research University College London Hospitals Biomedical Research Centre, University College London, London, UK
[6]Alan Turing Institute, London, UK
[7]Robertson Centre for Biostatistics and Clinical Trials, University of Glasgow, Glasgow, UK
[8]National Heart and Lung Institute, Imperial College, London, UK
[9]Department of Cardiology, Division Heart and Lungs, University Medical Center Utrecht, Utrecht University, Utrecht, The Netherlands
[10]Institute of Cardiovascular Science, Faculty of Population Health Sciences, University College London, London, UK

**Acknowledgements** This study was approved by the Medicines and Healthcare Products Regulatory Agency (UK) Independent Scientific Advisory Committee protocol reference: 17_015, under Section 251 (NHS Social Care Act 2006). This study is based in part on data from the Clinical Practice Research Datalink obtained under license from the UK Medicines and Healthcare products Regulatory Agency. The data are provided by patients and collected by the NHS as part of their care and support. The interpretation and conclusions contained in this study are those of the author(s) alone. Hospital Episode Statistics Copyright (2019) is reused with the permission of The Health & Social Care Information Centre. All rights reserved. This study was carried out as part of the CALIBER © resource (https://www.ucl.ac.uk/health-informatics/caliber and https://www.caliberresearch.org/). CALIBER, led from the UCL Institute of Health Informatics, is a research resource providing validated electronic health record phenotyping algorithms and tools for national structured data sources.

**Contributors** AU has designed the research, analysed and interpreted the data and drafted the manuscript. IV, AH, FWA and SK have designed the research, interpreted the data, critically revised the manuscript and supervised. AU, SD, HH, AS, JC and DG have interpreted the data and critically revised the manuscript.

**Funding** This study is part of the BigData@Heart program that has received funding from the Innovative Medicines Initiative 2 Joint Undertaking under grant agreement No 116074. This Joint Undertaking receives support from the European Union's Horizon 2020 research and innovation programme and EFPIA. This work was supported by Health Data Research UK [grant number N/A], which receives its funding from Health Data Research UK Ltd (NIWA1) funded by the UK Medical Research Council, Engineering and Physical Sciences Research Council, Economic and Social Research Council, Department of Health and Social Care (England), Chief Scientist Office of the Scottish Government Health and Social Care Directorates, Health and Social Care Research and Development Division (Welsh Government), Public Health Agency (Northern Ireland), British Heart Foundation and the Wellcome Trust. Work at the University College London Institute of Health Informatics and Institute of Cardiovascular Science is supported by a British Heart Foundation Accelerator Award (AA/18/6/24223). FA is supported by UCL Hospitals NIHR Biomedical Research Centre. IV is supported by a grant from the Dutch Heart Foundation [grant DHF project 'Facts and Figures']. SD is supported by an Alan Turing Fellowship. HH is supported by an NIHR Senior Investigator Award. ADS is funded by a post-doctoral fellowship from THIS Institute. JC received research grants from Bayer, Novartis and Vifor and honoraria for steering committees from Amgen, Bayer, Novartis and Servier.

**Competing interests** None declared.

**Patient consent for publication** Not required.

**Provenance and peer review** Not commissioned; externally peer reviewed.

**Data availability statement** All data were provided anonymised and are not publicly available due to their sensitive nature. Data may be obtained from the Clinical Practice Research Datalink (https://www.cprd.com). EHR phenotypes are available from the CALIBER resource (https://www.caliberresearch.org). The protocol may be obtained via the Clinical Practice Research Datalink under protocol reference: 17_015. No additional data is available.

peer-reviewed. Any opinions or recommendations discussed are solely those of the author(s) and are not endorsed by BMJ. BMJ disclaims all liability and responsibility arising from any reliance placed on the content. Where the content includes any translated material, BMJ does not warrant the accuracy and reliability of the translations (including but not limited to local regulations, clinical guidelines, terminology, drug names and drug dosages), and is not responsible for any error and/or omissions arising from translation and adaptation or otherwise.

**ORCID iDs**
Alicia Uijl http://orcid.org/0000-0003-2835-7741
Harry Hemingway http://orcid.org/0000-0003-2279-0624

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
