## [Reviewer comments · BMJ Open]

ARTICLE DETAILS

TITLE (PROVISIONAL)	Temporal Trends in Heart Failure Medication Prescription in a Population-Based Cohort Study
AUTHORS	Uijl, Alicia; Vaartjes, Ilonca; Denaxas, S; Hemingway, Harry; Shah, Anoop; Cleland, J; Grobbee, Diederick; Hoes, Arno; Asselbergs, Folkert; Koudstaal, Stefan

VERSION 1 – REVIEW

REVIEWER	WJ Kruik-Kollöffel Ziekenhuis Groep Twente
REVIEW RETURNED	03-Sep-2020

GENERAL COMMENTS	Dear editor, Thank you for the invitation to revise the manuscript entitled “Temporal Trends in Heart Failure Medication Use in a Population-Based Cohort Study” (bmjopen-2020-043290) submitted to your journal. The authors are to be commended for their effort to transform a lot of numbers into interesting Figures and Tables. However, I have some major concerns with the manuscript in this version, which might have to do with unclarity of the text. 1. Prescription data. The Clinical Practice Research Datalink (CPRD) consists of primary care HER, HES and ONS. Are all prescription data in this datalink? Medication prescribed by cardiologists, are they registered? Patients treated in out-patient HF clinics by cardiologists and specialized HF nurses, where are their prescriptions registered? If these prescription data are not in the dataset, the prescriptions are seriously underestimated, especially for patient hospitalized.2. Diagnosis recorded. What do the authors mean with “HF diagnosis recorded in primary care”. Is the diagnosis recorded in the files of the GP, or is the patient treated by the GP? A patient treated for HF by the cardiologist in the hospital might be excellently treated while the GP file does not contain the recording of HF.3. Primary versus secondary care. Patients treated in primary care are in general more stable and their HF is less severe, whereas patients treated in secondary care, especially as they have been hospitalized for HF are in worse condition. To combine those two groups is risky. To base the major conclusion of the manuscript (“improving communication between primary and secondary care”) on differences
--

between the two groups is even more risky. For example in the abstract “patients hospitalized for HF who had no follow-up in primary care”: what if patients had their follow-up in secondary care? Is HF care equal in primary and secondary care? The conclusions on page 15 lines 30-45 are much too certain in my opinion.

Minor concerns with references to page number on the top/line number of the pdf:

4. 2/3 (title): prescriptions are a proxy for dispensings -> dispensings are a proxy for use, this should be mentioned
5. 3/15 (70.2-85.7): is this the IQR?
6. 3/28 (RAS): the abbreviation is introduced without explanation
7. 3/50 (communication): see my major concerns, also in the rest of the manuscript
8. 4/15 (HFmrEF): the subphenotype HFmrEF is not mentioned in the manuscript
9. 5/25 (prior): “prior and after their diagnosis”, or “changes after diagnosis in relation to prior”, or....
10. 7/11 (however): change into “although”?
11. 7/35 (caliber): a reference to the website of CALIBER might be helpful (<https://www.ucl.ac.uk/health-informatics/caliber>)
12. 7/42 (registries): it is not clear why “heart failure disease registries” are mentioned here. For RCTs nobody needs explanation why you mention them.
13. 8/28 (index date): this is a bit confusing. Patient inclusion started January 1st 2002. Therefore a patient diagnoses with HF at January 2nd can be included. January 2nd being the index date. Data from 3 years prior to index date was included. Therefore CPRD data since January 2nd 1999 are required, is this right? However, the GP practice needed to have at least one year up-to-standard data recording?
14. 8/30 (READ): please explain, as it is a UK-NHS-only definition
15. 8/42 (up-to-standard): what is up-to-standard? Who judges?
16. 8/47 (de-registration): de-registration from the GP?
17. 8/47 (last data collection): which data? Prescriptions, or comorbidities as well?
18. 9/6 (starting with “Baseline patient...”): this sentence is unreadable and lacks a “)” sign
19. 9/13 (comorbidities): on what criterium is this selection of comorbidities based?
20. 9/50 (social deprivation and smoking): from which database are these data?
21. 9/50 (social deprivation): why is it added? The NHS after all takes care for alle UK-citizens equally?
22. 9/47 (ethnicity, smoking): these variables are only mentioned in Table 1. Why were they added?
23. 10/42 (Table 1, characteristics): a lot of the variables in this Table are only mentioned in this table. In het manuscript there is given no further meaning, interpretation, etc. So, why are they mentioned at all?

	24. 11/13 (Figures): the Figures are suitable to show a trend. However, if one reads the text in the Results paragraph, percentages would be useful. For example at the end of page 12 "After HF diagnosis, the most prescribed medication for women was a loop diuretic, while a higher proportion of men were prescribed a RAS-inhibitor." It is hard to find the exact numbers in all those lines and figures. I suggest you add some Figures and some statistics as well. For example only at time stamp 3 months since diagnosis. 25. 14/50 (room for improvement): it is crucial to start disease modifying drugs as soon as possible after hospitalization. Longer after diagnosis is too late. 26. 15/6 (if): the word "if" should be deleted? 27. 15/8 (diuretics primary care): the obvious explanation for less use of diuretics is lower frequency of symptoms and therefore less severe HF. 28. 15/25 (survival): if patients have less severe HF in primary care, better prognosis will be a logical consequence.... 29. 16/42 (elderly diuretics): no explanation is given for the higher prescription of diuretics in elderly patients. I am interested to hear the ideas of the investigators. This might be caused by more elderly patients on HFpEF. In HFpEF there are less possibilities for evidence-based treatment. Therefore, patients will be treated with symptom-control instead of disease modifying medications. Another explanation might be diagnoses in primary care versus secondary care: elderly more often diagnosed in primary care and GPs more reluctant to treat those patients too fierce. Or: elderly patients have more contraindications or intolerance to disease-modifying drugs. 30. 16/42 (women): no explanation is offered for the conclusion that "women received more loop diuretics." 31. 17/6 (limitations): this paragraph should be expanded with some of the points I mentioned above. Especially the missing information not only on the type of HF, but also any other information on the HF of the patient (ejection fraction, NTproBNP or even NYHA class) deserves more attention. 32. 17/38 (lowest prescription rates): of which drug(s)? 33. Figures: for patients entered at the end of the inclusion period only a minority might reach the time point of 3 months since diagnosis. How is dealt with this problem?
--	---

REVIEWER	Paulino Alvarez Cleveland Clinic, Ohio, United States
REVIEW RETURNED	06-Sep-2020

GENERAL COMMENTS	Thank you for the opportunity of reviewing the research paper from Dr.Uijl. "Temporal Trends in Heart Failure Medication Use in a Population-Based Cohort Study" . The quality of the data, analysis and graphic display are great. The interaction of primary care and heart failure management is important and often unrecognized. Comments: 1-Regarding the heart failure medications although the authors acknowledge the limitation of identifying the heart failure phenotype one possible way of trying to address this limitation is to perform a
--

	subgroup analysis in patients who were hospitalized (HES) limited to the patients who had an ICD coded related to systolic heart failure (ICD 9) or heart failure with reduced ejection fraction (ICD 10). 2- Another subgroup analysis to strengthen the conclusions is to evaluate the use of medications among patients with heart failure who had implantable defibrillators or cardiac resynchronization devices. 3- I would exclude patients with advanced renal disease CKD 4 or higher, cancer (excluding skin), and valvular disease such as aortic stenosis that are factors that may affect therapeutic decisions.
--	--

VERSION 1 – AUTHOR RESPONSE

Response to the reviewers:

Reviewer 1

Thank you for the invitation to revise the manuscript entitled “Temporal Trends in Heart Failure Medication Use in a Population-Based Cohort Study” (bmjopen-2020-043290) submitted to your journal. The authors are to be commended for their effort to transform a lot of numbers into interesting Figures and Tables. However, I have some major concerns with the manuscript in this version, which might have to do with unclarity of the text.

1. Prescription data. The Clinical Practice Research Datalink (CPRD) consists of primary care HER, HES and ONS. Are all prescription data in this datalink? Medication prescribed by cardiologists, are they registered? Patients treated in out-patient HF clinics by cardiologists and specialized HF nurses, where are their prescriptions registered? If these prescription data are not in the dataset, the prescriptions are seriously underestimated, especially for patient hospitalized.

We thank the reviewer for his/her remark. CPRD, HES and ONS are all separate datasets that have been linked with the NHS number. Prescription data is only available in CPRD, however in the UK the GP is involved in chronic disease management and therefore the vast majority of long term prescriptions are prescribed from primary care, even if they are initiated in the specialist inpatient or outpatient setting. CPRD includes all prescriptions from community pharmacies including treatments initiated by cardiologists or specialist nurses. Treatments administered during a hospital admission, such as intravenous inotropic agents, are not systematically recorded in GP records. Prescriptions from hospitals are usually only prescribed up to 2 weeks, after which the GP takes over regular prescribing. We have updated the manuscript to reflect this division better (page 8): *CPRD includes all prescriptions from community pharmacies. Prescriptions in CPRD were classified as:*

And in the limitations (page 16):

We only had medication prescription available in CPRD, not in HES. However, CPRD includes all prescriptions from general practice. Treatments administered during a hospital admission or discharge were not reported, such as intravenous inotropic agents.

2. Diagnosis recorded. What do the authors mean with “HF diagnosis recorded in primary care”. Is the diagnosis recorded in the files of the GP, or is the patient treated by the GP? A patient treated for HF by the cardiologist in the hospital might be excellently treated while the GP file does not contain the recording of HF.

The diagnosis recorded in primary care means that the GP has recorded that the patient has heart failure using a code from the Read V2 controlled clinical terminology. GPs are routinely informed of all admissions and outpatient consultations for the patients registered with them, and as part of good practice they are expected to record major diagnoses such as heart failure even if they do not make

the diagnosis themselves. Indeed, patients hospitalised for HF might be treated by a cardiologist. We have updated the methods in the manuscript (page 7):

In CPRD, events were defined by a diagnosis of HF based on Read (version 2) controlled clinical terminology codes (NHS coded clinical terms) ...

3. Primary versus secondary care. Patients treated in primary care are in general more stable and their HF is less severe, whereas patients treated in secondary care, especially as they have been hospitalized for HF are in worse condition. To combine those two groups is risky. To base the major conclusion of the manuscript (“improving communication between primary and secondary care”) on differences between the two groups is even more risky. For example in the abstract “patients hospitalized for HF who had no follow-up in primary care”: what if patients had their follow-up in secondary care? Is HF care equal in primary and secondary care? The conclusions on page 15 lines 30-45 are much too certain in my opinion. The reviewer raises an excellent point. We have updated the manuscript to reflect this uncertainty more (page 14):

In this study almost 40% of patients were not followed up in primary care after a HF hospitalisation. One reason could be that GPs do not formally register HF with a Read diagnostic code, but rather in free text descriptions. However, there could also be a potential quality of care gap between secondary and primary care, where patients are not treated optimally.

Minor concerns with references to page number on the top/line number of the pdf:

4. 2/3 (title): prescriptions are a proxy for dispensings -> dispensings are a proxy for use, this should be mentioned

We agree with the reviewer that the title might not reflect the content of the manuscript. We have updated the title to replace Use with Prescription (page 1):

Temporal Trends in Heart Failure Medication Prescription in a Population-Based Cohort Study.

5. 3/15 (70.2-85.7): is this the IQR?

Indeed, this reflects the IQR, we have updated the manuscript accordingly (page 2):

... 79.1 [interquartile range 70.2-85.7] ...

6. 3/28 (RAS): the abbreviation is introduced without explanation

We apologize for this mistake. We have updated the abstract accordingly (page 2):

...higher prescription rates of loop diuretics in women and elderly patients together with lower prescription rates of angiotensin converting enzyme-inhibitors and/or angiotensin II receptor blockers, beta-blockers, or MR-antagonists in these patients...

7. 3/50 (communication): see my major concerns, also in the rest of the manuscript

We have updated the manuscript at different places mentioned in question 3.

8. 4/15 (HFmrEF): the subphenotype HFmrEF is not mentioned in the manuscript

We have updated the manuscript to include HFmrEF (page 3 and page 12):

- *Unable to differentiate between HF subphenotypes HF with reduced, mid-range or preserved ejection fraction.*

We were unable to distinguish HF with reduced ejection fraction (HFrEF), HF with mid-range ejection fraction, and HF with preserved ejection fraction (HFpEF) based on medical records...

9. 5/25 (prior): “prior and after their diagnosis”, or “changes after diagnosis in relation to prior”, or....

The key questions have been removed as it was not required to add this page to the manuscript.

10. 7/11 (however): change into “although”?

We have revised the manuscript accordingly (page 6):

Once diagnosed, initiation and up titration of guideline recommended therapies can reduce morbidity and mortality, although 5-year survival still remains 20% to 50%.

11. 7/35 (caliber): a reference to the website of CALIBER might be helpful

(<https://www.ucl.ac.uk/health-informatics/caliber>)

We have referenced a CALIBER website in another part of the methods which describes the definitions of variables. References to the CALIBER platform were made in reference 9 and 10. We think these references adequately describe the CALIBER data resource and do not require a reference to the UCL CALIBER website.

12. 7/42 (registries): it is not clear why “heart failure disease registries” are mentioned here. For RCTs nobody needs explanation why you mention them.

We have removed section regarding heart failure disease registries and only kept RCTs (page 6):

Given the amount of information available, medication use of all HF patients in the community may be investigated – including those which are underrepresented in randomised clinical trials.

13. 8/28 (index date): this is a bit confusing. Patient inclusion started January 1st 2002. Therefore a patient diagnoses with HF at January 2nd can be included. January 2nd being the index date. Data from 3 years prior to index date was included. Therefore CPRD data since January 2nd 1999 are required, is this right? However, the GP practice needed to have at least one year up-to-standard data recording?

Indeed, the reviewer is correct here. Linked GP and hospital EHRs includes data from 1998 onwards. Therefore, the first HF record from January 1st 2002 is when patients could be included. The eligibility criteria are applied in every CALIBER study and are a quality control check, most patients are registered for much longer than 1 year with the same GP. We have updated the manuscript (page 7): *... in a practice that had at least one year of up-to-standard data recording in CPRD (data quality check).*

Data on EHR phenotyping variables from HF patients up to 3 years prior to index date were included in this study.

14. 8/30 (READ): please explain, as it is a UK-NHS-only definition

We have updated the methods in the manuscript (page 7):

In CPRD, events were defined by a diagnosis of HF based on Read (version 2) controlled clinical terminology codes (NHS coded clinical terms) ...

15. 8/42 (up-to-standard): what is up-to-standard? Who judges?

This is regulated by CPRD. They have two key methods of ensuring quality of the data made available to the research community: 1) the ‘acceptable research quality’ flag, a patient-level quality marker; 2) and the Up To Standard date, a practice-level quality marker. We have updated the manuscript (page 7):

... in a practice that had at least one year of up-to-standard data recording in CPRD (data quality check).

16. 8/47 (de-registration): de-registration from the GP?

Indeed, we have updated the manuscript (page 7):

Individuals were censored at the earliest date from the date of de-registration in CPRD...

17. 8/47 (last data collection): which data? Prescriptions, or comorbidities as well?

This part in the methods only describes the study population and inclusion of patients. The last data collection date refers to the last date that CPRD has provided information. The data in our database was 1998 to July 2016. Therefore, most practices would have updated information. However, it does happen that practices drop out from providing information to CPRD, which means we cannot follow patients after that day. We have updated the manuscript accordingly (page 7):

... *the last data collection date of a practice in CPRD* ...

18. 9/6 (starting with "Baseline patient..."): this sentence is unreadable and lacks a ")" sign. We have added the ")" sign and we added a break after medication prescription to make it more readable. We also changed some () to [] to better distinguish between groups of characteristics (page 8):

Baseline patient characteristics were based on records from CPRD and/or HES prior to index date, including demographics [age, sex, ethnicity, social deprivation] cardiovascular risk factors [smoking, BMI, diastolic blood pressure and systolic blood pressure and estimated glomerular filtration rate], comorbidities [a medical history of atrial fibrillation, chronic obstructive pulmonary disease (COPD), diabetes, hypertension, ischaemic heart disease, valvular disease and history of cancer] and medication prescription. Prescriptions were classified as: RAS-inhibitors (Angiotensin converting enzyme-inhibitors and/or angiotensin II receptor blockers), beta-blockers, mineralocorticoid receptor-antagonists (MR-antagonists) and loop diuretics.

19. 9/13 (comorbidities): on what criterium is this selection of comorbidities based?

We thank the reviewer for this question. Comorbidities were chosen based on importance and contribution to heart failure. Previously we have shown that AF, COPD, diabetes and hypertension are risk factors for heart failure: Uijl A, Koudstaal S, Direk K, et al. Risk factors for incident heart failure in age- and sex-specific strata: a population-based cohort using linked electronic health records. *Eur. J. Heart Fail.* 2019.

20. 9/50 (social deprivation and smoking): from which database are these data?

Social deprivation is an index of the primary care practice. Smoking is measured in CPRD. We have updated the manuscript accordingly (page 8-9):

Social deprivation was measured as quintiles of the index of multiple deprivation of the geographical area of the primary care practice, a score calculated based on seven indices of deprivation: income, employment, health and disability, education, barrier to housing and services, crime and living environment. (15) Smoking status in CPRD was classified as never, ex- or current smokers.

21. 9/50 (social deprivation): why is it added? The NHS after all takes care for alle UK-citizens equally?

Social-economic status is an important factor to take into account. We, and others, have previously shown that low social-economic status is associated with a higher incidence of heart failure. Patients with lower SES could have difficulty in finding access to care. We wanted to investigate whether this was visible in temporal trends. As we show, indeed, there are no differences in the lowest quintile of social deprivation vs. the rest).

22. 9/47 (ethnicity, smoking): these variables are only mentioned in Table 1. Why were they added?

These variables, and others in table 1, were added to provide an overview of the heart failure patient seen in clinical practice. As we have no information regarding HF subphenotypes, we provided as much information as possible regarding these patients.

23. 10/42 (Table 1, characteristics): a lot of the variables in this Table are only mentioned in this table. In het manuscript there is given no further meaning, interpretation, etc. So, why are they mentioned at all?

See the answer to question 22.

24. 11/13 (Figures): the Figures are suitable to show a trend. However, if one reads the text in the Results paragraph, percentages would be useful. For example at the end of page 12 “After HF diagnosis, the most prescribed medication for women was a loop diuretic, while a higher proportion of men were prescribed a RAS-inhibitor.” It is hard to find the exact numbers in all those lines and figures. I suggest you add some Figures and some statistics as well. For example only at time stamp 3 months since diagnosis. Exact percentages for prescription use at 3 months after HF diagnosis are included in Table 1.

25. 14/50 (room for improvement): it is crucial to start disease modifying drugs as soon as possible after hospitalization. Longer after diagnosis is too late. We thank the reviewer for this comment. We did not want to imply to start medication later rather than sooner. However, we do want to emphasise that physicians should not hesitate starting or up-titrating medication past the 6-month mark. We have revised this section (page 13-14):
This leaves room for improvement in starting treatment at any time point following a HF diagnosis, for example if patients hospitalised with acute HF do not immediately tolerate negative inotropic medication such as beta-blockers.

26. 15/6 (if): the word “if” should be deleted?
Indeed, if should be deleted here, we have updated the manuscript (page 14):
We found differences in prescription patterns between patients with a HF diagnosis recorded in different settings.

27. 15/8 (diuretics primary care): the obvious explanation for less use of diuretics is lower frequency of symptoms and therefore less severe HF. We have updated the manuscript accordingly (page 14):
It could be that these patients have less severe fluid overload or symptoms that requires alleviation by loop diuretics, and thus less severe HF.

28. 15/25 (survival): if patients have less severe HF in primary care, better prognosis will be a logical consequence.... We agree with the reviewer (page 14):
It could be that these patients have less severe fluid overload or symptoms that requires alleviation by loop diuretics, and thus less severe HF.

29. 16/42 (elderly diuretics): no explanation is given for the higher prescription of diuretics in elderly patients. I am interested to hear the ideas of the investigators. This might be caused by more elderly patients on HFpEF. In HFpEF there are less possibilities for evidence-based treatment. Therefore, patients will be treated with symptom-control instead of disease modifying medications. Another explanation might be diagnoses in primary care versus secondary care: elderly more often diagnosed in primary care and GPs more reluctant to treat those patients too fierce. Or: elderly patients have more contraindications or intolerance to disease-modifying drugs. The reviewer raises an excellent point. There are indeed many different possibilities as to why elderly (and women) might have higher prescriptions of diuretics. However, as we did not have detailed information on ejection fraction, NYHA class or NT-proBNP measurements, we refrained from making too many claims or assumptions that could be related to these measurements.

We described loop diuretics and HFpEF in elderly patients here (page 15):

This could potentially be related to the fact that elderly women are more likely to develop HFpEF and therefore tend to be treated symptomatically with loop diuretics, rather than with RAS-inhibitors and beta-blockers.

and updated the manuscript here (page 16) with contraindications in elderly:

However, elderly patients might have more contraindications or intolerances to RAS-inhibitors, beta-blockers and MR-antagonists and might therefore be more often treated with loop diuretics for symptom control.

30. 16/42 (women): no explanation is offered for the conclusion that "women received more loop diuretics."

We describe that women were slightly older than men and that elderly women are more likely to develop HFpEF (page 15):

This could potentially be related to the fact that elderly women are more likely to develop HFpEF and therefore tend to be treated symptomatically with loop diuretics, rather than with RAS-inhibitors and beta-blockers.

31. 17/6 (limitations): this paragraph should be expanded with some of the points I mentioned above. Especially the missing information not only on the type of HF, but also any other information on the HF of the patient (ejection fraction, NTproBNP or even NYHA class) deserves more attention.

We have updated the limitations (page 16):

Nor did we have information on NYHA class or NT-proBNP biomarker levels.

32. 17/38 (lowest prescription rates): of which drug(s)?

We have updated the manuscript (page 16):

Large differences were observed between HF patient groups, with lowest prescription rates of RAS-inhibitors, beta-blockers and MR-antagonists in women, elderly patients, and those without a primary care diagnosis.

33. Figures: for patients entered at the end of the inclusion period only a minority might reach the time point of 3 months since diagnosis. How is dealt with this problem?

We have taken this into account by not including the last months of data we had available to us, data was available from 1998 to July 2016. By only including patients to the end of 2015 we were able to include prescription data up to 3 months after HF diagnosis if a patients was diagnosed end of 2015.

Reviewer 2

Thank you for the opportunity of reviewing the research paper from Dr. Uijl. "Temporal Trends in Heart Failure Medication Use in a Population-Based Cohort Study". The quality of the data, analysis and graphic display are great. The interaction of primary care and heart failure management is important and often unrecognized.

Comments:

1-Regarding the heart failure medications although the authors acknowledge the limitation of identifying the heart failure phenotype one possible way of trying to address this limitation is to perform a subgroup analysis in patients who were hospitalized (HES) limited to the patients who had an ICD coded related to systolic heart failure (ICD 9) or heart failure with reduced ejection fraction (ICD 10).

We thank the reviewer for this suggestion. The 'I50.2 Heart failure with reduced ejection fraction' code is ICD-10 CM, not the WHO ICD-10 which used in the UK. The ICD-10 code I50 is unspecific for HFrEF or HFpEF.

2- Another subgroup analysis to strengthen the conclusions is to evaluate the use of medications among patients with heart failure who had implantable defibrillators or cardiac re synchronization devices.

Indeed, we agree with the reviewer this would be an interesting subgroup analysis. However, due to the low numbers this would not be a representative subgroup and it would be difficult to draw conclusions based on this.

3- I would exclude patients with advanced renal disease CKD 4 or higher, cancer (excluding skin), and valvular disease such as aortic stenosis that are factors that may affect therapeutic decisions. We thank the reviewer for this comment. However, we aimed to describe heart failure medication trends in a real-world population, we believe that if we exclude these patients, the medication prescription trends would not reflect an unselected real-world population.

VERSION 2 – REVIEW

REVIEWER	WJ Kruijk-Kolloffel Ziekenhuisgroep Twente (Hospital Group Twente)
REVIEW RETURNED	06-Dec-2020

GENERAL COMMENTS	Thank you for you revised manuscript. In my opinion, the quality of the paper has improved and it is more informative than the previous version. I have a response on some of the points. I kept the numbers of my original response: 1. Prescription data. The authors response for page 8:  up to 2 weeks, after which the GP takes over regular prescribing. we have updated the manuscript to reflect this division better (page 8): CPRD includes all prescriptions from community pharmacies. Prescriptions in CPRD were classified as:  However, the text in the manuscript is:  valvular disease and history of cancer)] and medication prescription. CPRD includes all prescriptions from the general practice. Prescriptions in CPRD were classified as: RAS-inhibitors (Angiotensin converting enzyme-inhibitors and/or angiotensin II  I suppose the text in the response is the right one? Please adjust the text in the manuscript accordingly. The text on page 16 now: Furthermore, we only had medication prescription available in primary care, not in hospital care. However, CPRD includes all prescriptions from community. Treatments administered during a hospital admission or discharge were not reported, such as intravenous inotropic agents. I suggest the following text, if I understand the authors correctly: Treatments administered during a hospital admission or discharge were not reported, such as intravenous inotropic agents. CPRD includes all prescriptions to non-hospitalized patients from community pharmacies.
---

	2. Diagnosis recorded. Abstract-results, your text: ...patients hospitalised for HF who had no follow-up in primary care.... I suggest: ...patients hospitalised for HF who had no recorded follow-up in primary care... 3. Primary versus secondary care. Excellent revision. 22/23. Variables in Table 1. The authors responded: “These variables, and others in table 1, were added to provide an overview of the heart failure patient seen in clinical practice. As we have no information regarding HF subphenotypes, we provided as much information as possible regarding these patients.” Mentioning all those variables, however, raises the question: is this group of patients representative for HF patients in general? I suppose that is why this information is shown? Could you tell us this in 1 or 2 sentences and preferentially an adequate reference? 24. Figures This is not what I meant, but I will drop this point. 29/30. Diuretics in elderly and women I thank the authors as they provide the reader with some reasonable explanations.
--	---

REVIEWER	Paulino Alvarez Cleveland Clinic, OH, United States.
REVIEW RETURNED	22-Nov-2020

GENERAL COMMENTS	The main limitation of this report is that there are no strategies to identify or differentiate patients with reduced and preserved ejection fraction. This issue is particularly important because for patients with preserved ejection fraction there is no specific pharmacological treatment that has shown improvement in outcomes. Heart failure with preserved ejection fraction is more prevalent in the elderly and women, the subgroups mentioned by the authors that have the lowest prescription of RAS inhibitors, beta blockers and MR antagonist. Because of this it is difficult to propose a potential improvement strategy when the problem has not been characterized. Given that as mentioned by the authors in their response the characterization of the type of heart failure is not possible, the interpretation of the conclusion of this manuscript is challenging.
--

VERSION 2 – AUTHOR RESPONSE

Reviewer 1

Comments to the Author

Thank you for your revised manuscript. In my opinion, the quality of the paper has improved and it is more informative than the previous version.

I have a response on some of the points. I kept the numbers of my original response:

1. Prescription data.

The authors response for page 8: *Community pharmacies*

However, the text in the manuscript is: *general practice*

I suppose the text in the response is the right one? Please adjust the text in the manuscript accordingly.

We thank the Reviewer for pointing out this inconsistency. We are very sorry the text in the response did not match the manuscript. We decided to use general practice instead of community pharmacies, as the prescriptions are distributed through community pharmacies, however prescriptions are from the general practice. Second reason is for consistency; we did not want to introduce another term. General practice in the revised manuscript is thus correct.

The text on page 16 now:

Furthermore, we only had medication prescription available in primary care, not in hospital care. However, CPRD includes all prescriptions from community. Treatments administered during a hospital admission or discharge were not reported, such as intravenous inotropic agents.

I suggest the following text, if I understand the authors correctly:

Treatments administered during a hospital admission or dispensed at discharge from hospital pharmacies were not reported. CPRD only includes prescriptions from community pharmacies. We have included the suggestion from Reviewer 1 in the text, we replaced community pharmacies with general practice for consistency reasons. Please find the updated text here:

Furthermore, treatments administered during a hospital admission or discharge were not reported, such as intravenous inotropic agents. However, CPRD includes all prescriptions from general practice to non-hospitalized patients.

2. Diagnosis recorded.

Abstract-results, your text:

...patients hospitalised for HF who had no follow-up in primary care....

I suggest:

...patients hospitalised for HF who had no recorded follow-up in primary care...

We have included this suggestion in the abstract:

and lastly, patients hospitalised for HF who had no recorded follow-up in primary care had considerably lower prescription rates compared to patients with a HF diagnosis in primary care with or without HF hospitalisation.

3. Primary versus secondary care.

Excellent revision.

We would like to thank the Reviewer for their suggestions that resulted in this revision.

22/23. Variables in Table 1.

The authors responded:

“These variables, and others in table 1, were added to provide an overview of the heart failure patient seen in clinical practice. As we have no information regarding HF subphenotypes, we provided as much information as possible regarding these patients.”

Mentioning all those variables, however, raises the question: is this group of patients representative for HF patients in general? I suppose that is why this information is shown? Could you tell us this in 1 or 2 sentences and preferentially an adequate reference?

The data from CPRD and HES is representative of the UK population; previous studies have shown the validity of the diagnostic codes used. The patients included in this analysis with a diagnosis of heart failure made either in primary or secondary care, reflect the experience in general practice,

which might differ from the patients enrolled in RCTs, who tend to be younger and predominantly men.

Reference to the representativeness and validity of diagnostic codes has been made in the methods (page 6):

Previous work has shown that these patients are representative of the general population in the UK. (11, 12)

24. Figures

This is not what I meant, but I will drop this point.

We would like to thank the Reviewer for dropping this point.

29/30. Diuretics in elderly and women

I thank the authors as they provide the reader with some reasonable explanations.

We would like to thank the Reviewer for their suggestions that resulted in this revision.

Reviewer 2

Comments to the Author

The main limitation of this report is that there are no strategies to identify or differentiate patients with reduced and preserved ejection fraction. This issue is particularly important because for patients with preserved ejection fraction there is no specific pharmacological treatment that has shown improvement in outcomes. Heart failure with preserved ejection fraction is more prevalent in the elderly and women, the subgroups mentioned by the authors that have the lowest prescription of RAS inhibitors, beta blockers and MR antagonist. Because of this it is difficult to propose a potential improvement strategy when the problem has not been characterized. Given that as mentioned by the authors in their response the characterization of the type of heart failure is not possible, the interpretation of the conclusion of this manuscript is challenging.

The Reviewer raises a good point. Guideline adherence on pharmacological treatment for heart failure with reduced ejection fraction is important and observational HF registries (SwedeHF, CHAMP-HF, ASIAN-HF and many other datasets) can provide the data needed to draw inferences on this research question.

We agree we have not used data which can address that research question. However, as stated in the title, our aim was on temporal trends in pharmacological treatment of HF in the general population, not only over time (2002 to 2015) but also following an index diagnosis of HF. How the absence of ejection fraction influences the interpretation of the presented data is discussed in the discussion section rather than the limitations. We agree the interpretation of conclusions is challenging, but feel that the revised manuscript is balanced and provides readers the ability to draw their own conclusions.

To minimise confusion, we have critically revised the conclusion of the abstract and removed the suggested improvements in quality of care, which could be interpreted as being related to guideline adherence.

Updated abstract (page 2):

For most patients, few changes were seen in pharmacological management of HF in the six months following diagnosis.

Updated conclusion (page 16):

Most changes in prescriptions occurred within 6 months prior to or 6 months following a diagnosis of HF, with little change thereafter, suggesting further opportunities to improve HF management..

VERSION 3 – REVIEW

REVIEWER	Paulino Alvarez Cleveland Clinic, Ohio, United States
REVIEW RETURNED	04-Feb-2021
GENERAL COMMENTS	My comments have been addressed especially the change in the conclusion of the study. No further observations. Well done.